# A Powdered Simulant of Triacetone Triperoxide (TATP) for Safe Testing of X-ray Transmission Screening Equipment

**DOI:** 10.3390/molecules25061473

**Published:** 2020-03-24

**Authors:** Mitja Vahčič, David Anderson, John Seghers, Hanne Leys, Miguel Ruiz Oses, Grzegorz Rarata, Maximino Fernández García, Rosana Prados Román, Daniel Pellico Escudero

**Affiliations:** 1European Commission, Joint Research Centre, Retieseweg 111, 2440 Geel, Belgium; mitja.vahcic@ec.europa.eu (M.V.); John.seghers@ec.europa.eu (J.S.); Hanne.leys@ec.europa.eu (H.L.); Miguel.RUIZ-OSES@ec.europa.eu (M.R.O.); Grzegorz.RARATA@ec.europa.eu (G.R.); 2Instituto Nacional de Técnica Aeroespacial, Madrid, Ctra. Ajalvir km.4, 28850 Torrejón de Ardoz, Spain; fernandezgma@inta.es (M.F.G.); pradosrr@inta.es (R.P.R.); 3Ingeniería de Sistemas para la Defensa de España S.A. A Spanish Public Enterprise 100% belonging to INTA. Calle Beatriz de Bobadilla, 3, 28040 Madrid, Spain; pellicoed.pers_externo@inta.es

**Keywords:** TATP, simulants, X-ray, explosives detection, aviation security, testing, harmonisation, standardisation

## Abstract

Explosives detection systems (EDS) based on X-ray are used at airports to screen baggage for the presence of explosives. Once EDS are installed in airports, however, it can be challenging to test the EDS equipment and verify that it continues to perform at the highest level, because of the impracticality of introducing bulk explosives into civil aviation airports. The problem is particularly acute for sensitive homemade explosives, such as triacetone triperoxide (TATP). This paper describes our work to develop a safe, accurate and stable simulant for TATP for EDS based on X-ray transmission. Bulk quantities of TATP were synthesised and characterised especially for this project, and we describe the unique challenges and safety considerations of collecting this data. Our calculations show that the expanded measurement uncertainty with a coverage factor of k = 2 is 5.7% for bulk density and 1.0% for *Zeff* at 24 months.

## 1. Introduction

In Europe, civil aviation security is regulated at EU level under Framework Regulation 300/2008 [1] and its related supplementing and implementing legislation [2]. The detailed measures for the implementation of these basic standards are laid down in the Commission Implementing Regulation (EU) 2015/1998 [3]. This legislation includes minimum performance requirements for security screening equipment, including explosives detection systems used to screen baggage for the presence of explosives.

In aviation security, explosives detection systems (EDS) is a specific term that refers to X-ray equipment with the functionality to automatically detect (and indicate by means of an alarm) the presence of explosives in screened baggage. EU legislation requires that all baggage has to be screened before being loaded onto the aircraft hold. Although multiple screening solutions are permitted by legislation, the sheer number of bags and the limited time available for screening mean that in practice almost all items of hold baggage are screened by EDS. In Europe and the United States, EDS equipment is tested extensively by specialist test centres [4,5] prior to approval for operational use in airports. Once EDS are approved and installed in airports, however, there are limited means to routinely test the EDS equipment and verify that it continues to perform at the highest level, because of the impracticality of introducing bulk explosives into civil aviation airports. The situation is particularly acute for sensitive homemade explosives, such as triacetone triperoxide (TATP).

TATP was discovered in 1895 by Richard Wolffenstein [6]. It belongs to a class of ketone peroxides, which can be found in many industrial applications such as initiators for polymerisation reactions in the production of silicone or polyester resins [7]. TATP is produced illicitly as a homemade explosive, due to the availability of precursors in household products, and it has been used in recent terrorist incidents in Europe and worldwide. TATP has the chemical formula C_9_H_18_O_6_ and its molecule is sketched in Figure 1.

The purpose of this work was to develop high-quality explosives simulants for TATP that can be used as standardised testing materials to verify the detection performance of EDS. The simulants had to be inert, non-toxic and mimic the relevant properties of explosives. For transmission X-ray systems, those properties are bulk density and effective atomic number, *Zeff*. It was also considered important that—just like actual TATP—the simulant is in a powder form.

This paper describes the development of TATP powder bulk simulants for EDS carried out at the European Commission’s Joint Research Centre (JRC) in Geel, Belgium. The target parameters were determined from highly specialised measurements of several hundred grams of TATP that were synthesised and characterised especially for this project by the Instituto Nacional de Técnica Aeroespacial (INTA), in Spain. We describe the unique challenges and safety considerations of collecting such characterisation data. To the best of our knowledge, our characterisation data of bulk quantities of TATP using dual-energy, computer tomography (DECT) equipment is the first such data collected in Europe. The project was safely executed, and the synthesis and characterisation steps were successful. However, the measures required to minimise the hazards of working with sensitive primary explosives in large quantities illustrate the need amongst the testing community for safe alternatives. We conclude that our simulants described in this paper are fit for purpose for the safe testing of the detection of TATP by explosives detection systems based on X-ray transmission.

Finally, we mention that this paper complements a recent paper by the authors on the development of solid, polymer-bonded simulants of other explosives [8], in which we describe in more detail the context of our work, the principle of operation of explosives detection systems, and the calibration and long-term stability of the equipment used. We have chosen not to duplicate those sections in this paper.

## 2. Results and Discussion

### 2.1. Production of TATP Simulants

We identified two main formulations for producing TATP simulants. Initial formulations were based on vanillin, caffeine, malic acid and sodium bicarbonate. Although the simulants based on these formulations had the required values of bulk density and effective atomic number, they suffered from a couple of drawbacks. The most obvious issue was the strong vanilla odour, which could be somewhat mitigated by tightly sealed containers. More unsatisfactory, however, was the unexpected tendency of the vanillin part of the simulant to partly recrystallize and go from free-flowing powder to a semi-hard powdered matrix. This issue could be partly reversed by input of mechanical energy to the simulant.

A second formulation was conceived based on stearates, caffeine and benzoates. The stearate-based formulations did not suffer from the drawbacks of the vanillin-based formulations, so we decided to focus on only the stearate formulations for further optimisation. The advantage of simulants based on stearates is that the amount of stearate present in the simulant and the level of compression of the mixture can be varied to manipulate the bulk density of the resulting simulant. Photos of the stearate-based TATP simulants are shown in Figure 2.

### 2.2. Accuracy of TATP Simulants

The accuracy of our stearate-based simulants of TATP was determined by comparing their density and effective atomic number to that of real TATP, as measured on the same equipment. Characterisation data was acquired using DECT X-ray equipment. This equipment has passed ECAC testing for so-called standard EDS-C2 [4]. The measurements of TATP were performed at INTA under contract to JRC and are described in more detail in the next section. The data consists of 15 screenings to determine average values of *Zeff* and bulk mass density, and this data provided us with target parameters to develop a TATP simulant. The sample of real TATP measured at INTA using DECT equipment had a bulk density of 0.66 g/cm^3^ and an effective atomic number, *Zeff*, of 7.10. The repeatability of the TATP measurements in terms of relative standard deviation (RSD) was 0.6% and 0.1% for density and *Zeff* respectively. The accuracy of six different TATP simulants relative to the measured values of real TATP is indicated in Figure 3.

We can see that the *Zeff* data exhibit very little variation across the simulants, and the measurements are very repeatable. Expressed as relative standard deviation (RSD), intermediate precision for *Zeff* was 0.3%. This confirms the efficacy of the mixing and homogenisation, as all six simulants were produced from the same batch of ingredients.

The density data exhibits greater variability, which reflects both the variation in the final density of the simulants, and the increased uncertainty inherent in measurements of powder densities using X-ray transmission equipment. Expressed as relative standard deviation (RSD), intermediate precision for bulk mass density was 1.5%.

### 2.3. Stability of Simulants

Stability testing is necessary to establish conditions for storage as well as conditions for dispatch to customers. A stability study was carried out using a standard design model where samples are measured at various time points and the linear regression analysis is used at the end to calculate the stability of the samples. The measurements at each time point were performed under repeatability conditions, and the data is shown in Figure 4.

To determine the stability of the simulants, we followed the volume integrity of the simulants along with bulk density and *Zeff*. These parameters were systematically followed for nine months at room temperature (21 °C). Stabilities at lower and elevated temperatures were not tested. Volume integrity stability testing of powder simulants shows no significant self-compaction over the test period. In Figure 3, we can see that four of the six simulants indicate a slight increase in density during the first 3 months, presumably due to settling, after which they were stable. Stability calculations indicate with a confidence level of 95% that the simulants are stable to within ± 2% at 24 months for bulk mass density and ± 0.3% for *Zeff*. These results show acceptable long-term stability of the simulants.

### 2.4. Measurement Uncertainty

To estimate the measurement uncertainty, we followed the Guide to the Expression of Uncertainty in Measurement [9]. We identified four main contributors associated with combined measurement uncertainty: homogeneity, intermediate precision, long-term stability and long-term instrument stability. Our calculations show that the expanded measurement uncertainty with a coverage factor of k = 2 is 5.7% for bulk density and 1.0% for *Zeff* at 24 months.

## 3. Materials and Methods

### 3.1. Explosives Detection Systems (EDS)

The density and *Zeff* of TATP and our simulants were determined using commercial explosives detection systems (EDS), namely the XT2080SI Kylin (Nuctech Company Limited, Beijing, China), with a nominal energy of 160 keV. This EDS has passed testing by the European Civil Aviation Conference [4]. In general, EDS equipment is designed to indicate the presence/absence of explosives inside scanned baggage, but not for quantitative measurements of material properties. Furthermore, the data processing and detection algorithms are trade secrets. For the EDS equipment used in this work, the manufacturer made available a software tool to extract values of density and effective number for the scanned materials. The calibration and long-term stability of the equipment was determined using high-purity reference materials and the results have been reported [8] in a previous paper by the authors on polymer-bonded simulants.

### 3.2. TATP Synthesis and Characterisation

A bulk amount of real TATP (around 400 g) was synthesised and subsequently characterised using DECT equipment. This part of the work was co-ordinated and carried out by INTA at the facilities in Cuadros, Leon (Spain). To the best of our knowledge, this was the first time ever that bulk amounts of TATP were characterised with DECT equipment in Europe. The data collection of bulk TATP samples with the EDS equipment required several phases: synthesis, transport, handling, testing and destruction. There was no readily available information on safety protocols for handling and transporting hundreds of grams of dry bulk samples, nor information on how to test these samples with DECT equipment. Thus, a risk assessment was carried out that led to the development of specific safety protocols before the start of the project. All phases were only performed by highly qualified personnel, under the use of the appropriate safety measures. Strategies were established in all the phases to minimise risks from electrostatic discharge, friction, impact, temperature, etc. During all steps involving bulk TATP, a medical team and ambulance were present. Work-up tests were carried out before the main test in order to train personnel in the various tasks to be performed, minimising the risk of possible errors and identifying improvements to the protocols.

#### 3.2.1. In Situ Synthesis

For safety reasons, the synthesis of the TATP was carried out as close as possible to the testing bunker where the EDS was located. For this reason, INTA personnel specialized in the synthesis of energetic materials were deployed to INTA’s facilities in Leon. The test site at Leon did not have any synthesis laboratory, so one of the buildings was conditioned to accommodate an in situ laboratory with all the safety measures in place, as well as all the necessary instrumentation and equipment.

There are numerous references in the literature on the synthesis of TATP [10,11,12]. Almost all of them refer, however, to small-scale synthesis (i.e., a few grams) used to investigate its properties. In this work, hundreds of grams of TATP were synthetized in two batches. The synthesis procedure used in this work was adapted from the procedures in the literature [11] with some modifications to improve safety, considering the larger quantities involved. The synthesis route is summarised in Figure 5. A mixture of acetone and cooled hydrogen peroxide was used. Sulphuric acid was added while controlling the temperature and speed of addition at all times. Stirring of the mixture continued for some time under controlled conditions, and then stirring was switched off and the reaction was left to stand. The precipitate obtained (see Figure 6a) was filtered and washed, and the resulting TATP was dried and placed in a suitable container.

#### 3.2.2. Handling and Transport

For handling and transport, all relevant safety measures were provided to the personnel (face shields, anti-static gowns and shoes, gloves) and EOD (explosive ordnance disposal) or military protection equipment (i.e., suits for EOD operators, helmets with and without visors, vests, etc.). The handling and transport of the synthesized sample was carried out by military personnel with EOD qualifications by INTA (see Figure 6b). The TATP had to be transported from the synthesis laboratory to the bunker (approximately 3 km), and from the bunker to the blasting area (approximately 400 m). Measures were taken to fix the TATP container to the vehicle as well as to protect the driver (EOD personnel). Among other things, insulating material and a small powder magazine were used to transport the TATP sample.

#### 3.2.3. Characterisation Testing

There are numerous published studies on the detection of TATP with various technologies with bulk, diluted, trace, or vapour TATP samples. In those studies, the detection is performed on very small amounts (less than one gram) of TATP [13,14,15,16,17,18,19,20]. Publications on testing with EDS were also reviewed; however, no references could be found on how to perform tests with this type of detection equipment with hundreds of grams of bulk TATP samples.

Testing comprised 15 sequential screenings of the TATP using DECT equipment (Nuctech XT2080SI Kylin) installed inside a bunker at INTA’s facilities in Leon, Spain. High-purity reference materials were also scanned before and after the measurements of TATP, for calibration and stability testing purposes. The TATP samples had to be placed in trays and these placed at the entrance to the EDS for analysis. Measures were taken in order to improve the entry and exit of the trays with the TATP, and a trolley was built for the placement and handling of the TATP in the trays. Once outside the measuring tunnel, they had to be collected. This process was carried out manually by the EOD personnel. The manufacturer was consulted on the possible effects that the EDS system could have on the TATP explosive. An improvised control centre was deployed outside the bunker and the EDS had to be adapted to allow it to be controlled from a greater distance than usual. From the control centre, the work of the different teams was co-ordinated, and the EDS equipment was operated. A remote video monitoring system was used to verify in real time the correct execution of the test inside the bunker. Once the runs were completed, the data generated by the system was analysed to verify the successful acquisition of data, thus marking the completion of the testing phase.

#### 3.2.4. Destruction

There are different methods of destroying TATP [21,22,23,24,25,26]: detonation, TATP decomposition (thermal, acid based, catalytic) and dilution and burning. Our requirements were that the TATP had to be destroyed safely as soon as the EDS testing was completed, and that its total destruction had to be ensured. Accordingly, the destruction of the bulk quantity of TATP was carried out by controlled detonation in a blasting field very close to the EDS test site. A special wooden frame was constructed to house the sample and a detonation cord filled with pentaerythritol tetranitrate (PETN) was used to connect the TATP with the electric detonator. The detonation was manually initiated by EOD personnel from a nearby blasting control building. A high-speed camera (FASTCAM-APX RS PHOTRON Color) was used to record the detonation. For safety reasons, the camera settings required for correct recording were made before the sample arrived in the area. The recording was controlled remotely from the blasting control building. Once the TATP was in place, the detonation and recording began. Just before the moment of detonation, the outdoor light conditions changed from overcast to sunny, resulting in some saturated highlights in the video footage. Some still images of the detonation sequence are shown in Figure 7.

Disclaimer: Cyclic acetone peroxides are sensitive to accidental detonation from impact, friction, electric discharge and flame. The synthesis, transportation, handling and measurement of TATP described in this paper are dangerous operations that should only be performed by highly qualified personnel using the strictest safety precautions for handling primary explosives. Neither the European Commission nor the Instituto Nacional de Técnica Aeroespacial shall be responsible for any damage, loss, harm of any kind suffered by third parties attempting to replicate the work described in this paper.

### 3.3. Selection of Simulant Ingredients

Effective atomic number is a critical concept for the automated detection of explosives using dual-energy x-ray equipment. Along with density, these two parameters are the primary features used by software algorithms to detect explosives in scanned baggage. However, effective atomic number is a (semi-) empirical parameter, and there are multiple definitions in literature [27]. In this work, we adopted a definition and algorithm for determining effective atomic number developed by the Lawrence Livermore National Laboratory [28,29]. This definition, referred to as *Ze* (to differentiate from *Zeff*) has several advantages, namely that it is based on a physical model instead of an empirical one, that it comprises one fixed algorithm over a range of materials and energy spectra of interest, and that each value of the parameter corresponds to a well-defined x-ray absorption behaviour. At the heart of the Ze approach is the calculation of energy-dependent photon cross-sections *σ(Ex)*, based on the sum of the cross sections of the constituent atoms in compounds and mixtures, weighted according to the relative electron fraction, *a_i_*:$$\sigma_{e}\left(mixture,\;Ex\right)=\overset{N}{\underset{i=1}{\sum }}a_{i}\sigma_{e}\left(Z_{i},\;Ex\right)$$
where *a_i_* is given by:$$a_{i}=\frac{n_{i}Z_{i}}{\sum_{j=1}^{N}n_{j}Z_{j}}$$
and *n_i_* is the number atoms, *Z_i_* is the atomic number for each element, *i*, in the mixture, and *N* is the number of constituent elements in the mixture.

We developed our own implementation [30] using Microsoft Excel 2010 to calculate *Ze*. We used this software to screen substances for their potential use as simulants ingredients. Ingredients were systematically screened and checked based on their *Zeff* values, C:H:N:O ratios, potential toxicity and aggregate state, and the necessary proportions of ingredients to achieve the required effective atomic number of the resulting mixture were determined.

### 3.4. Materials

Chemicals used for synthesis of TATP explosive used to obtain reference values were concentrated sulphuric acid (Sigma Aldrich, Darmstadt, Germany), hydrogen peroxide at 30% (Sigma Aldrich, Darmstadt, Germany) synthesis-grade acetone (Sigma Aldrich, Darmstadt Germany) and sodium bicarbonate (Sigma Aldrich, Darmstadt, Germany). Chemicals used for preparation and development of simulant formulations were as follows: vanillin, caffeine, lithium stearate, sodium stearate, calcium stearate (Alfa Aesar, Kandel, Germany), malic acid, glucose, sorbitol, xylitol, ascorbic acid, cyanuric acid (Sigma Aldrich, Darmstadt, Germany), sodium bicarbonate, sodium benzoate, acetylsalicylic acid (VWR, Leuven, Belgium), 1,6 hexanediol and TRIS (Merck, Darmstadt, Germany). All simulants described in this paper were in powder form and were initially contained in plastic bags, but in order to gain better control over powder volume packaging was changed to cylindrical PVC transparent tubes with a diameter of 70 ± 0.1 mm. The height of the cylindrical containers and the size of the bags depended on the mass of the simulant. For long-term stability studies and calibration purposes of EDS, pure polyoxymethylene (Delrin, Conventry, United Kingdom), polytetrafluoroethylene (PTFE), graphite, silicon and aluminium rods with the diameter of 60 mm and length of 200 mm, from American Elements (Los Angeles, CA, USA) were used.

### 3.5. Equipment for Simulant Production

The Turbula^®^ benchtop 3D mixer (WAB, Basel, Switzerland with a screwcap polypropylene container was used to homogenize initial powder mixtures. In later stages of development, the powders were homogenized using a 3D mixer (DynaMIX CM200 (WAB, Basel, Switzerland)). For drying of the mixture slurries, a temperature-controlled drying cabinet was used (Heraeus Series 6000/UT6760 Thermo Scientific, Dreieich, Germany). For grinding of the powders, a cryo-grinding vibrating mill was used (KHD Humboldt Wedag, Cologne, Germany). For compacting of the powder mixtures, a vibratory feeder Laborette 24 was used (Fritsch GmbH, Idar-Oberstein, Germany).

### 3.6. Material Processing for Simulant Production

The process of bulk powder explosive simulant production required that the simulant components were weighed according to calculated proportions and dissolved in deionized water to generate a slurry, which was then homogenized by mixing. The slurry was dried in a temperature-controlled oven. The mixture was then frozen overnight in liquid nitrogen in a stainless-steel container and subsequently milled using a cryo-grinding vibrating mill. The mill was maintained below −120 °C throughout the process. This was a necessary because milling at room temperature would cause melting due to friction created during the milling process. Milling served two goals. Firstly, it ensured all particles were approximately the same average size in order to prevent stratification during mixing and compression. Secondly, milling controls the bulk volume and electrostatic interaction between particles, which in turn affect the bulk density of the resulting simulant. The smaller the particles, the bigger the electrostatic forces between them. The mixture was then slowly brought to room temperature in a dry atmosphere (this was achieved by displacing air with dry nitrogen gas) in order to avoid water condensation on the powder. This prevented the contamination of the simulant by water, which can change the density, *Zeff*, and physical properties of the material. The powder mixture was then mixed in a 3D mixer for one hour to homogenize the distribution of the different powder fractions. Finally, the simulant powder mixture was packed into a PVC tube of 70 × 500 mm, and a thickness of 15 µm, and vibrated to the desired volume. The excess of tube material was cut and the ends were sealed, to ensure the bulk density remained matched with the target one.

## 4. Conclusions

Explosives detection systems (EDS) play a key role in aviation security, and they represent the last line of defence in preventing explosives from being brought on board aircraft in baggage. There is a need to verify the detection performance of EDS during their operational lifetime, but working with explosives in an operational environment is challenging. The situation is particularly acute for sensitive homemade explosives like TATP.

We acquired characterization data of bulk density and effective atomic number of several hundred grams of TATP using dual-energy, computer-tomography EDS. The challenges and safety precautions employed to minimise the risk of this kind of work are described in this paper. We used the characterization data to develop simulants for TATP based on a formulation comprising stearates, caffeine and benzoates. Compared to solid simulants, controlling and the bulk density of powders is more challenging. We described the material processing steps used to produce our simulants.

To the best of our knowledge, our material is the first one that was designed using actual measurements of bulk amounts (i.e., around 400 g) of TATP explosive, and the material is accompanied by reliable test data. Our experience is that commercial TATP simulants are not supplied with much, or even any, information on accuracy of density or *Zeff*, composition, or stability, and hence do not have sufficient quality control or quality assurance to be used during formal testing in aviation security. Based on the comparison of bulk density and effective atomic number of our simulants with the real TATP sample measured with the same equipment, we conclude that our simulants are fit for purpose as a safe alternative for verifying the detection of TATP in baggage screening equipment based on X-ray transmission. The process described in this paper is adaptable, and additional TATP simulants can be produced with different properties if deemed appropriate, for example, in the light of intelligence on specific recipes.

## 5. Patents

A patent linked to this work has been filed with the European Patent Office in 2019.

## Figures and Tables

**Figure 1 molecules-25-01473-f001:**
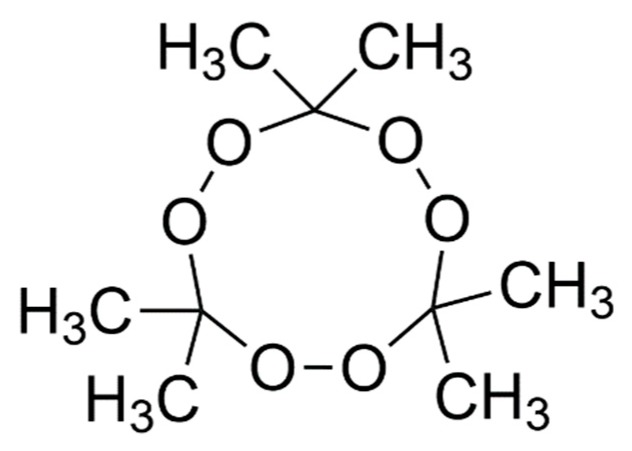
Molecular structure of triacetone triperoxide (TATP), C_9_H_18_O_6_.

**Figure 2 molecules-25-01473-f002:**
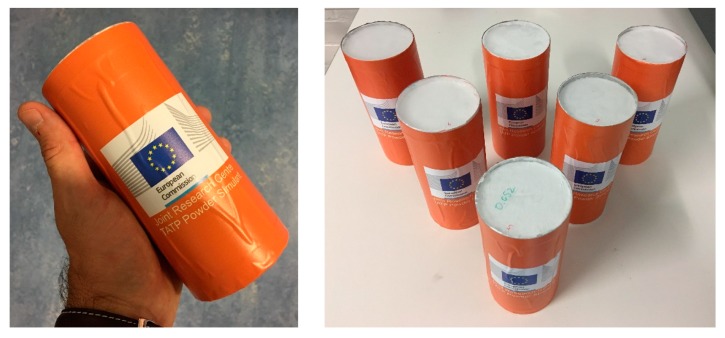
Photos of our TATP simulant based on stearates, caffeine and benzoates.

**Figure 3 molecules-25-01473-f003:**
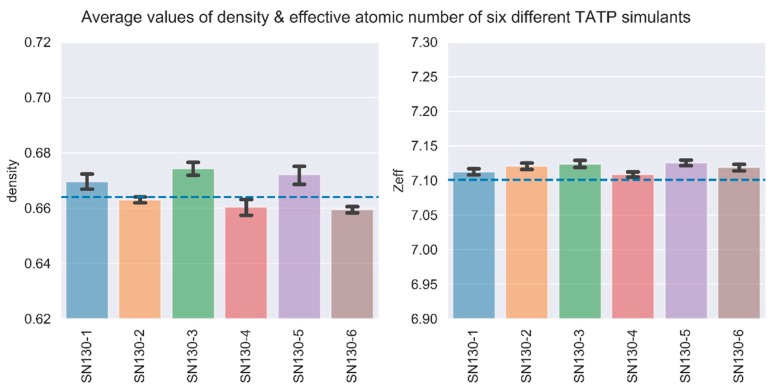
Density and effective atomic number of six TATP simulants. The bars indicate the mean of 60 runs per simulant (over four different days), and the error bars indicate the 95% confidence intervals. The dashed lines show the target values of the real TATP: density = 0.66, *Zeff* = 7.10.

**Figure 4 molecules-25-01473-f004:**
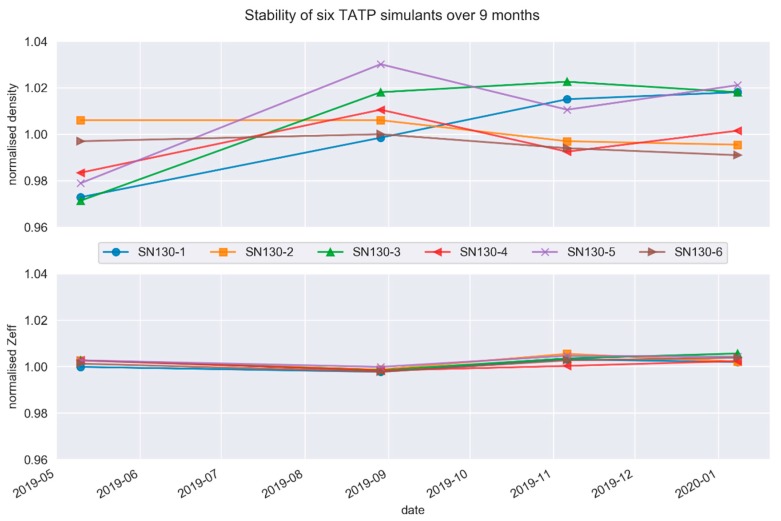
Stability testing data for six simulants for bulk density (**top**) and effective atomic number, *Zeff* (**bottom**) recorded at room temperature (21 °C) over a period of nine months.

**Figure 5 molecules-25-01473-f005:**
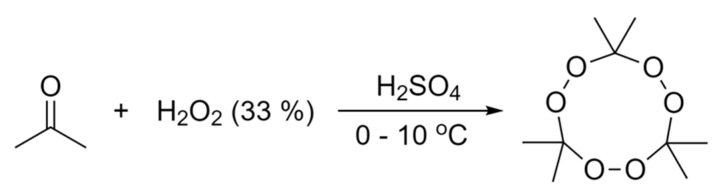
TATP synthesis route used in this study.

**Figure 6 molecules-25-01473-f006:**
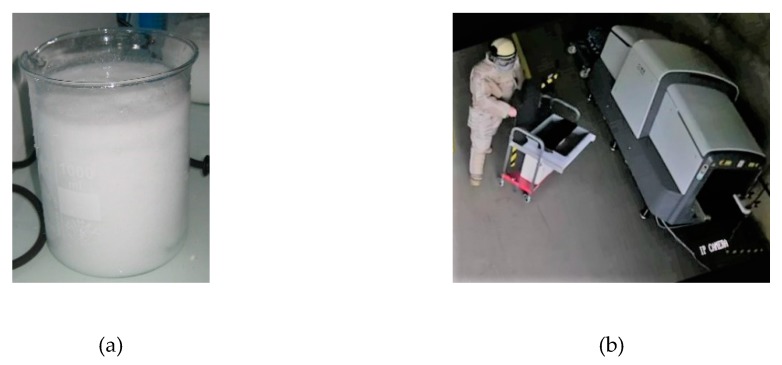
(**a**) Precipitated TATP and (**b**) TATP handling by EOD-qualified military personnel.

**Figure 7 molecules-25-01473-f007:**
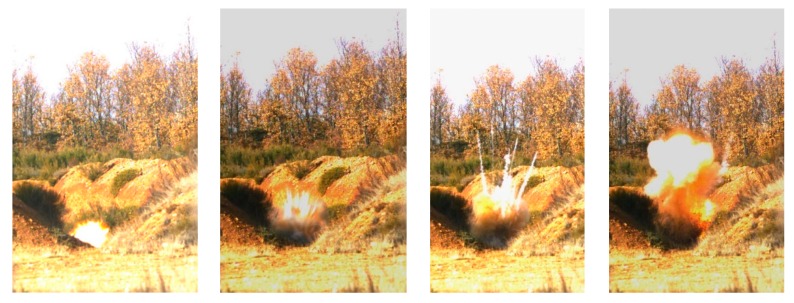
Detonation sequence of bulk TATP recorded with a high-speed camera at the INTA test site near Leon, Spain.

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
