# Peer review of "A Powdered Simulant of Triacetone Triperoxide (TATP) for Safe Testing of X-ray Transmission Screening Equipment"

_molecules, 2020, doi:10.3390/molecules25061473_

Round 1

Reviewer 1 Report

The manuscript could be published by Molecules after major revision. There are many issues that the authors should take care of prior to its publication. The manuscript does not specify according to which density standards the authors packed their TATP. This explosive could be packed in various densities. The manuscript does not specify polymorphism and crystal size of the TATP that the authors prepared and analyzed. The manuscript does not specify the energy levels that were used to measure the Zeff. A professional in this field would ask to see not normalized values. Also, the authors should specify the type of machine, type of the material used for container (and according to which standard this container was used) and TATP’s packing density in the container. The authors should be aware that there are several known TATP’s simulants (for x-ray) and they should clearly indicate what are main advantages of their material.

Reviewer 2 Report

This paper 
describes an important work to develop a safe, accurate and stable simulant (based on a formulation comprising stearates, 
caffeine and benzoates) for TATP for EDS based on X- 
ray transmission. Authors synthesised  bulk quantities of TATP and acquired characterization data of bulk density and effective atomic number of several 
hundred grams of TATP using dual-energy, computer tomography EDS. Based on the comparison of bulk density and effective atomic number of the simulants with the 
real TATP sample measured with the same equipment, they concluded that the simulants are fit for 
purpose as a safe alternative for verifying the detection of TATP in baggage screening equipment based on X-ray transmission.
 Expanded measurement uncertainty (with a coverage factor of k = 2) is 5.7 
% for bulk density and 1.0 % for Zeff at 24 months is a really promising result.  Authors described the unique challenges and safety considerations of collecting this data. Material processing steps used to produce the simulants are described. 
The paper will be interesting to the readers from a wider scientific background.

However, the paper needs couple of important minor revisions.

a) Please mention the formula of TATP and a sketch of the molecule.

b) It is recommended to provide some chemical analysis (XRD, NMR, DSC) of the synthesised TATP sample. How authors concluded that they made a pure sample of TATP without having any other impurities (such as DADP or unreacted starting material). Is there any difference in purity, yield compared to small-scale synthesis ?

c) In the synthesis section it is necessary to provide a scheme of TATP synthesis (detailed synthesis can be referenced). Readers should extract all important information from the paper.

d) Few of the references are not easily accessible (for example Ref 30). Section 3.3 -- Selection of simulant ingredients-- needs to be improved. How authors used the software to screen substances for their potential use as simulants ingredients, and calculated the necessary proportions of ingredients to achieve the required effective atomic number of the mixture is not clear. 

e) In conclusion, the paper needs to be stand-alone. It can be seriously improved by including couple of missing/incomplete information- such as how effective atomic number are calculated for a mixture. Perhaps an equation helps. Lastly, how the results of hundreds of grams of TATP compare with couple of grams of TATP? 

Round 2

Reviewer 1 Report

The authors made revision of their manuscript and now it is better suited for its publication. Prior to the final approval, the authors should take into account two following publications:

  1. "Design and validation of inert homemade explosive simulants for X-ray-based inspection systems"; Faust, Anthony A.; Nacson, Sabatino; Koffler, Bruce; Bourbeau, Eric; Gagne, Louis; Laing, Robin; Anderson, C. John; Proceedings of SPIE (2014), 9073(Chemical, Biological, Radiological, Nuclear, and Explosives (CBRNE) Sensing XV), 90730V/1-90730V/12; DOI:10.1117/12.2058035.
  2. "TATP ray-level simulation explosive simulant"; Lu, Xiujuan; Xu, Wenguo; Liu, Jiping; Kan, Meixiu; Faming Zhuanli Shenqing (2012), CN 102838436 A 20121226. Abstract: The title TATP (triacetone triperoxide) ray-level simulation explosive simulant comprises three formulas: formula 1 is prepd. from 1,6-hexanediol and malonic acid at a mol ratio 1:1, formula 2 is prepd. from 1,6-hexanediol, linoleic acid, citric acid (monohydrate) at a wt. ratio 37.8530:10.0790:52.067, and formula 3 is prepd. from 1,6-hexanediol, α-type oxalic acid and hydroquinone at a wt. ratio 46.856:36.68:3.1877. The invention can replace TATP explosives for calibration of security app. and instruments, and avoids security risks.